# Effect of mode of healthcare delivery on job satisfaction and intention to quit among nurses in Canada during the COVID-19 pandemic

**Safoura Zangiabadi**[1⊙]**, Hossam Ali-Hassan** [2⊙] *

**1** School of Kinesiology and Health Sciences, Keele campus, York University, Toronto, Canada,
**2** Department of International Studies, Glendon campus, York University, Toronto, Canada

⊙ These authors contributed equally to this work.
* hossama@yorku.ca

**Data Availability Statement:** Data and survey details from Statistics Canada can be found at: https://www23.statcan.gc.ca/imdb/p2SV.pl?

## Abstract

The COVID-19 pandemic resulted in a major shift in the delivery of healthcare services with the adoption of care modalities to address the diverse needs of patients. Besides, nurses, the largest profession in the healthcare sector, were imposed with challenges caused by the pandemic that influenced their intention to leave their profession. The aim of the study was to examine the influence of mode of healthcare delivery on nurses' intention to quit job due to lack of satisfaction during the pandemic in Canada. This cross-sectional study utilized data from the Health Care Workers' Experiences During the Pandemic (SHCWEP) survey, conducted by Statistics Canada, that targeted healthcare workers aged 18 and over who resided in the ten provinces of Canada during the COVID-19 pandemic. The main outcome of the study was nurses' intention to quit within two years due to lack of job satisfaction. The mode of healthcare delivery was categorized into; in-person, online, or blended. Multivariable logistic regression was performed to examine the association between mode of healthcare delivery and intention to quit job after adjusting for sociodemographic, job-, and health-related factors. Analysis for the present study was restricted to 3,430 nurses, weighted to represent 353,980 Canadian nurses. Intention to quit job, within the next two years, due to lack of satisfaction was reported by 16.4% of the nurses. Results showed that when compared to participants who provided in-person healthcare services, those who delivered online or blended healthcare services were at decreased odds of intention to quit their job due to lack of job satisfaction (OR = 0.47, 95% CI: 0.43–0.50 and OR = 0.64, 95% CI: 0.61–0.67, respectively). Findings from this study can inform interventions and policy reforms to address nurses' needs and provide organizational support to enhance their retention and improve patient care during times of crisis.

Function=getSurvey&SDDS=5362 https://www150.statcan.gc.ca/n1/en/catalogue/13250006.

**Funding:** The authors received no specific funding for this work.

**Competing interests:** The authors have declared that no competing interests exist.

## Introduction

The declaration of the coronavirus disease 2019 (COVID-19) pandemic by the World Health Organization (WHO) in March 2020 led to a radical transformation in the delivery of healthcare services worldwide [1]. As a result, healthcare organizations adopted the use of virtual care to reduce the risk of viral transmission in clinical settings and to deliver optimal care for patients [2,3]. Meanwhile, most provincial healthcare systems in Canada responded to the pandemic by transitioning to the use of virtual care services ranging from telephone to video-conferencing and text messaging besides essential in-person care [4]. According to Statistics Canada, in the fall of 2021, 24% of healthcare professionals provided virtual care services since the pandemic started. Among those who delivered virtual care, 87% used the phone for providing care, followed by 47% through video, and 26% via email, text, or instant messaging [5]. Virtual care allowed for efficient triage of patients, rapid access to health information, and greater work flexibility for healthcare providers [6]. Despite its positive implications, the integration of virtual care into workflows has not been without challenges. Previous studies reported that healthcare providers faced usability and workflow issues that posed difficulties in incorporating virtual care into their practice compared to in-person care [7–9]. For instance, lack of proper training on virtual medical software platforms contributed to healthcare providers' reluctancy to use virtual care [10]. Similarly, virtual care did not accommodate healthcare providers' need to conduct direct physical examinations on patients, limiting the care they could provide [6,7].

Moreover, during this time, nurses were among the frontline healthcare workers who provided direct patient care and were subjected to unprecedented pressure from various stressors, including increased workload, greater exposure to the virus, isolation from family members, and lack of personal protective equipment (PPE) [11,12]. Of consequence, nurses exhibited a higher prevalence of anxiety and depression than other healthcare providers [12–14]. The added physical and psychological strain of the pandemic impacted nurses' intention to consider leaving their current positions [15,16]. A meta-analysis of ten research studies reported that 31.7% of nurses intended to leave their job during the COVID-19 pandemic [17]. In Ontario, 83% of registered nurses expressed that their mental health was negatively impacted during the pandemic and 34% considered leaving their profession due to the declined mental health and burnout [18]. In addition, a recent study of nursing staff in Quebec showed that nurses who reported self-infection with COVID-19 or infection of team members had a greater intention to leave the profession [19]. Although numerous studies, since the onset of the pandemic, have examined the influence of COVID-19 on intention to leave the profession, few studies have shown that nurses' intention to leave their job maybe influenced by job satisfaction and the quality of work environment [20]. A descriptive study of 371 nurses actively working during the pandemic in Turkey showed that as nurses experienced higher levels of burnout, their occupational satisfaction declined, and the intention to leave their nursing profession increased [21]. Lastly, pay satisfaction affects nurses' turnover intention and is important in maintaining strong ties with healthcare organizations and staying in the profession [22]. For instance, findings of a recent survey indicated that 60% of registered practical nurses (RPNs) in Ontario consider leaving the profession, while 75% expressed that a raise in wages would make them more likely to remain in their job [23].

Health frameworks are necessary to address the needs of nursing staff and provide support and resources to increase their retention and ultimately enhance the quality of patient care during these challenging times. Yet to date, no study has examined the association between the mode of healthcare delivery and the nurses' intention to leave job due to lack of satisfaction. Studying the association between the mode of healthcare delivery and nurses' intention to leave their job due to lack of satisfaction is crucial as it can reveal how different healthcare

delivery models impact nurse job satisfaction, which, in turn, affects workforce retention. Understanding this link is essential for healthcare organizations to adopt effective delivery models that support nurse satisfaction and ultimately ensure the availability of a stable and dedicated nursing workforce, vital for providing high-quality patient care. Hence, it is essential to address the job satisfaction of nursing staff as they are an integral part of the healthcare system that provide high quality care to patients. Thereby, understanding the perceived experiences of nurses on the care delivery modality is important to find the right balance in the use of mode of delivery to ensure their satisfaction. This paper aims to assess the effect of the mode of healthcare delivery on job satisfaction among nurses during the COVID-19 pandemic in Canada. The research question assesses the association between mode of healthcare delivery, including in-person, online, or blended on the intention to quit within two years due to the lack of job satisfaction among nurses during the COVID-19 pandemic in Canada.

## Materials and methods

This study utilized data from the Health Care Workers' Experiences During the Pandemic (SHCWEP) survey conducted by Statistics Canada between September to November 2021 [24]. The SHCWEP was a cross-sectional survey that collected information on the impact of the COVID-19 pandemic on healthcare workers in Canada. The target population included healthcare workers aged 18 and over who worked in a healthcare setting since the start of the COVID-19 pandemic and resided in the ten provinces of Canada. A total of 22,293 were estimated to have met the eligibility criteria, out of which 12,246 individuals completed the survey with a response rate of 54.9% [24]. The survey was conducted via electronic questionnaire or computer-assisted telephone interviewing and participation was voluntary. The present study was restricted to nurses who completed the (SHCWEP) survey.

The main outcome of the study was nurses' intention to quit due to lack of job satisfaction. Information about this variable was collected based on the question, "How long are you planning to stay in your current job?". Respondents were provided with different options listed as: "Less than 6 months", "6 months to less than a year", "1 to 2 years", "3 to 5 years", and "6 or more years". Those who intended to quit their job within two years were further asked about the reason, with the question being, "What are the reasons that you might consider leaving or changing your job?". The answer choices included: "Retiring", "Job stress or burnout", "Lack of job satisfaction", "Concerns about your physical health and safety", "Concerns about the physical and mental health of household members or other close to you", "Financial impacts or concerns", "Long-term impacts of COVID-19 on healthcare system, including changes in method of delivery of health care", "Other career opportunity", and "Other". The responses to the answer choice of intent to quit because of lack of job satisfaction were recoded and assessed as a dichotomous variable, with individuals who want to quit due to lack of job satisfaction as "Yes" and those who intend to quit for other reasons than lack of job satisfaction or do not want to quit at all as "No".

The main independent variable of this study was mode of delivery of health care services during the COVID-19 pandemic, which was assessed by the question, "Since March 2020, how did you provide health care services to patients or clients at your primary job location?". Responses to the question were: "Over the phone", "Video meeting", "Email, text, or instant messaging", "In-person", "Other". The response categories were combined into three categories; in-person, online (combining video meeting, email, text, or instant messaging) and blended (combining in-person and online).

Covariates for the present study included several job-related factors (years worked in the job, number of locations worked at, type of healthcare job, received formal training on

Infection Prevention and Control (IPC), sociodemographic factors (gender, age, province of residence, household income, immigration status), and health-related factors (perceived general health, perceived mental health).

Descriptive statistics of the intention and reasons for quitting job within two years among nurses were conducted. Chi-square test and bivariate logistic regression models were performed to assess the relationship between each of the job-, sociodemographic-, and health-related factors with intention to quit job due to lack of satisfaction among nurses. In addition, one multivariable logistic regression model was conducted with the outcome being the intention to quit job due to lack of satisfaction and the main independent variable being mode of delivery of health care services adjusting for all the job-, sociodemographic-, and health-related variables. Adjusted Odds Ratio (ORs) and 95% Confidence Intervals were reported for the final model. Population weights were applied to each calculated estimate and bootstrapping was performed to adjust for the complex sampling methodology (reference of the survey). All analyses were conducted using the Statistical Package for the Social Sciences (SPSS, version 28.0). Statistical significance for all analyses was set to alpha of 0.05.

### Ethics statement

This study did not require ethics review by the York University Ethics Board as SHCWEP public use microdata files produced by Statistics Canada are publicly accessible [25] and appropriately protected by law via the Data Liberation Initiative [26].

## Results

Of the 3,638 nurses who were interviewed, 65 (1.8%) and 153 (4.2%) nurses were excluded as they did not provide healthcare services to patients and had missing information on their intention to quit job, respectively. Hence, analysis for the present study was restricted to 3,430, weighted to represent 353,980 Canadian nurses. Table 1 presents descriptive statistics of nurses' intentions and reasons for quitting a job within two years.

**Table 1. Intention and reasons of nurses in Canada for quitting current job within two years.**

| Factor | Number* | (%)# |
|---|---|---|
| **Intention to quit current job within 2 years** | | |
| Yes | 118,627 | 33.5 |
| No | 235,352 | 66.5 |
| **Reasons for quitting job#** | | |
| Retiring | 43,255 | 12.2 |
| Stress or burnout | 66,710 | 18.8 |
| Lack of job satisfaction | 47,611 | 13.5 |
| Concerns physical health | 35,957 | 10.2 |
| Concerns mental health | 57,977 | 16.4 |
| Concerns about household members | 21,321 | 6.0 |
| Financial impacts | 9,425 | 2.7 |
| Health care system | 35,908 | 10.1 |
| Other career opportunity | 32,943 | 9.3 |
| Other | 13,607 | 3.8 |

*Sample size is estimated using population weights.

#Percentages do not add to 100% because of multiple answers.

Of the respondents, 33.5% reported having the intention to quit their current job within the next two years, with reasons ranging from stress or burnout (18.8%) to concerns about mental health (16.4%) to lack of job satisfaction (13.5%), the main dependent variable for the present study. Among respondents, 89.8% were females, and more than 30% of the participants were 18 to 34 years old and resided in Ontario. The majority of participants (85.7%) provided in-person healthcare services, while 3.6% and 10.6% offered online and blended services to patients, respectively (Table 2).

Table 2 shows the results of the multivariable logistic regression model. Mode of delivery of health care services was significantly associated with intention to quit due to lack of job satisfaction. The results showed that when compared to participants who provided in-person healthcare services, those who delivered online or blended healthcare services were at decreased odds of intention to quit their job due to lack of job satisfaction (OR = 0.47, 95% CI: 0.43–0.50 and OR = 0.64, 95% CI: 0.61–0.67, respectively). Among the other job-related factors, those who worked less than 10 years were at increased odds for intention to quit due to lack of job satisfaction (OR = 1.88, 95% CI: 1.80–1.97) than those who worked 20 years or more. Similarly, participants who worked at multiple locations were at increased odds for intention to quit job for lack of satisfaction than those who worked at one location only (OR = 1.38, 95% CI: 1.35–1.42). Furthermore, the intention to quit due to lack of job satisfaction was significantly lower among those who worked in outpatient and ambulatory care compared to individuals who worked in acute care and significantly higher among those who did not receive formal training in Infection Prevention and Control (IPC) compared to those who received training.

Regarding sociodemographic-related factors, intention to quit due to lack of job satisfaction was significantly lower among females (OR = 0.95, 95% CI: 0.91–1.00) and older individuals who were 55 years and older compared to younger individuals (OR = 0.78, 95% CI: 0.75–0.82). Participants who lived in British Colombia were 1.62 times at increased odds for intention to quit for lack of job satisfaction than those who lived in Quebec (95% CI: 1.56–1.68). On the contrary, individuals with immigrant or non-permanent resident immigration status had significantly lower odds of intention for quitting for lack of job satisfaction than non-immigrants (OR = 0.51, 95% CI: 0.50–0.53). Lastly, with regards to health-related factors, those with self-perceived general and mental health ratings of very good and excellent expressed significantly decreased likelihood of quitting their job compared to participants with poor or fair self-perceived general and mental health (OR = 0.44, 95% CI = 0.43–0.46 and OR = 0.41, 95% CI: 0.40–0.43, respectively).

## Discussion

The emergence of the COVID-19 pandemic resulted in a major paradigm shift in delivery of healthcare services with the adoption of a virtual care model to meet the needs of patients without forfeiting the quality of patient care [27]. Moreover, the pandemic greatly impacted nurses' well-being and psychological health caused by excessive workload and high risk of infection, to provide optimal care, putting them at greater risk of leaving their job [28]. The present study examined the influence of the mode of healthcare delivery on nurses' intention to leave job due to lack of satisfaction during the COVID-19 pandemic in Canada. Intention to quit current job within the next two years was reported by 33.5% of the nurses, and 13.5% reported lack of job satisfaction as the reason behind intention to leave their profession. The results of this study demonstrated that nurses who provided online or blended healthcare services were significantly less likely to intend to quit due to lack of job satisfaction compared to nurses who provided in-person healthcare services during the COVID-19 pandemic. The findings of this

**Table 2. Characteristics of participants and relationships between socio-demographic, job and health-related factors and quitting due to lack of job satisfaction and among nurses in Canada.**

| Factor | %* | OR | 95%CI | p-value | Adjusted OR | 95%CI | p-value |
|---|---|---|---|---|---|---|---|
| **Job-related factors** | | | | | | | |
| **Mode of healthcare delivery** | | | | | | | |
| In-person | 85.7 | 1 | | | 1 | | |
| Online | 3.6 | 0.74 | 0.70, 0.78 | <0.001 | 0.47 | 0.43, 0.50 | <0.001 |
| Blended | 10.6 | 0.53 | 0.51, 0.55 | <0.001 | 0.64 | 0.61, 0.67 | <0.001 |
| **Years worked in the job** | | | | | | | |
| Less than 10 years | 42.5 | 3.06 | 2.97, 3.14 | <0.001 | 1.88 | 1.80, 1.97 | <0.001 |
| 10 to 19 years | 25.6 | 1.48 | 1.43, 1.53 | <0.001 | 1.73 | 1.66, 1.81 | <0.001 |
| 20 years or more | 25.6 | 1 | | | 1 | | |
| Other[#] | 6.4 | 1.94 | 1.85, 2.03 | <0.001 | 1.79 | 1.69, 1.90 | <0.001 |
| **Number of locations worked at** | | | | | | | |
| One location | 72.0 | 1 | | | 1 | | |
| Multiple locations | 25.1 | 1.21 | 1.18, 1.23 | <0.001 | 1.38 | 1.35, 1.42 | <0.001 |
| **Type of healthcare job location** | | | | | | | |
| Acute care | 61.8 | 1 | | | 1 | | |
| Long-term care | 17.3 | 1.03 | 1.01, 1.06 | 0.009 | 1.04 | 1.01, 1.08 | 0.005 |
| Outpatient and ambulatory care | 11.7 | 0.40 | 0.39, 0.42 | <0.001 | 0.41 | 0.39, 0.43 | <0.001 |
| Other[#] | 7.1 | 1.05 | 1.01, 1.09 | 0.01 | 1.13 | 1.08, 1.18 | <0.001 |
| **Received formal training on IPC** | | | | | | | |
| Yes | 90.7 | 1 | | | 1 | | |
| No/ Workplace do not have an IPC policy/valid skip | 8.7 | 2.78 | 2.71, 2.86 | <0.001 | 3.80 | 3.67, 3.93 | <0.001 |
| **Socio-demographic factors** | | | | | | | |
| **Gender** | | | | | | | |
| Male | 10.1 | 1 | | | 1 | | |
| Female | 89.8 | 0.82 | 0.79, 0.84 | <0.001 | 0.95 | 0.91, 1.00 | 0.03 |
| **Age** | | | | | | | |
| 18 to 34 years | 32.0 | 1 | | | 1 | | |
| 35 to 44 years | 24.8 | 0.38 | 0.37, 0.39 | <0.001 | 0.38 | 0.37, 0.40 | <0.001 |
| 45 to 54 years | 21.5 | 0.27 | 0.26, 0.27 | <0.001 | 0.47 | 0.45, 0.49 | <0.001 |
| 55 years and older | 21.5 | 0.34 | 0.33, 0.35 | <0.001 | 0.78 | 0.75, 0.82 | <0.001 |
| **Province of residence** | | | | | | | |
| Quebec | 22.9 | 1 | | | 1 | | |
| Atlantic provinces | 8.5 | 1.20 | 1.15, 1.25 | <0.001 | 0.86 | 0.82, 0.90 | <0.001 |
| Ontario | 36.6 | 1.33 | 1.29, 1.37 | <0.001 | 0.86 | 0.83, 0.88 | <0.001 |
| Manitoba | 3.9 | 1.26 | 1.20, 1.33 | <0.001 | 1.22 | 1.15, 1.30 | <0.001 |
| Saskatchewan | 3.2 | 1.18 | 1.12, 1.26 | <0.001 | 0.86 | 0.81, 0.92 | <0.001 |
| Alberta | 12.4 | 1.10 | 1.06, 1.14 | <0.001 | 0.88 | 0.85, 0.92 | <0.001 |
| British Columbia | 12.4 | 1.93 | 1.87, 1.99 | <0.001 | 1.62 | 1.56, 1.68 | <0.001 |
| **Household income** | | | | | | | |
| Under $100,000 | 29.5 | 1.69 | 1.65, 1.74 | <0.001 | 1.21 | 1.18, 1.25 | <0.001 |
| $100,000 to $149,999 | 26.7 | 0.97 | 0.95, 1.00 | 0.05 | 0.83 | 0.81, 0.86 | <0.001 |
| $150,000 or more | 28.3 | 1 | | | 1 | | |
| Not stated | 15.5 | 1.15 | 1.11, 1.19 | <0.001 | 0.99 | 0.95, 1.03 | 0.45 |
| **Immigration status** | | | | | | | |
| Non-immigrant | 74.8 | 1 | | | 1 | | |
| Immigrant or non-permanent resident | 22.5 | 0.53 | 0.51, 0.54 | <0.001 | 0.51 | 0.50, 0.53 | <0.001 |

*(Continued)*

**Table 2.** (Continued)

| Factor | %* | OR | 95%CI | p-value | Adjusted OR | 95%CI | p-value |
|---|---|---|---|---|---|---|---|
| **Health-related factors** | | | | | | | |
| **Perceived general health** | | | | | | | |
| Poor/Fair | 12.5 | 1 | | | 1 | | |
| Good | 36.7 | 0.46 | 0.44, 0.47 | <0.001 | 0.56 | 0.55, 0.58 | <0.001 |
| Very good/Excellent/ | 50.7 | 0.31 | 0.30, 0.32 | <0.001 | 0.44 | 0.43, 0.46 | <0.001 |
| **Perceived mental health** | | | | | | | |
| Poor/Fair | 24.4 | 1 | | | 1 | | |
| Good | 39.6 | 0.54 | 0.52, 0.55 | <0.001 | 0.79 | 0.77, 0.81 | <0.001 |
| Very good/Excellent | 35.8 | 0.23 | 0.23, 0.24 | <0.001 | 0.41 | 0.40, 0.43 | <0.001 |

* Percentages may not add to 100 because of missing data.

study highlight the importance of implementing health policies that provide support and resources for nurses to increase their retention rate and subsequently improve the quality of patient care.

Findings from this study reflect previous literature on physicians' positive attitudes toward using virtual care and higher job satisfaction [29–31]. A recent study of female healthcare workers in the United States demonstrated that employees who had the ability to work remotely from home during the pandemic reported less stress and higher job satisfaction. Additionally, results from the same study revealed that the flexibility of working remotely and the use of telemedicine allowed healthcare employees to feel safer while maintaining work efficiency and performance [29]. Alternative reasons for favorable perception of virtual care delivery could be attributed to greater efficiencies in the care process, increased work productivity, and consequently improved quality of life, which possibly influences job satisfaction [6,8]. A study of physicians in Saudi Arabia reported that the use of telemedicine was beneficial for their practice as it enabled them to provide care for patients who seldom visited the hospital and save commute time, allowing them to accomplish tasks more quickly and efficiently [32]. Another factor attributed to increased interest in the use of virtual healthcare services has been linked to its cost-effectiveness in delivering care that provides flexibility in scheduling and saving time for physicians, ultimately improving the quality of life of physicians [3]. Hence, telemedicine allows for flexible work arrangements that reduce burnout and improve work-life balance leading to higher job satisfaction and perhaps less turnover intention.

Among other job-related factors, years worked in the job, the number of locations worked at, type of healthcare job location, and receiving formal training on IPC were all significantly associated with the intention to leave job. Nurses who worked in their job for less than ten years and worked at multiple locations were more likely to quit than those who worked for more years in their job or worked at one location, respectively. These results are consistent with previous studies that showed that nurses with less work experience were more likely to leave their jobs [33,34]. This is explained as those with more work experience feel more acquainted with the hospital and consider it their second home [35]. Another explanation is that nursing is considered a highly stressful and complex profession in a high-risk workplace that could generate negative emotions and affect novice nurses' physical and mental health. Thereby, nurses with less professional experience and those with low levels of emotional resilience and professional skills are less likely to stay in a hospital environment [33]. Likewise, working at multiple locations could possibly further add to the emotional exhaustion and

burnout of nurses' experience given the stressful nature of their work paired with the uncertainty of the pandemic, leading to a higher intention to quit job [36].

Moreover, similar to previous studies [37,38], those who were not provided with pandemic-specific IPC training had a significantly higher intention to leave their job. The likely reason is that lack of appropriate IPC training mitigates safe workplace practices, increases the personal health risk of nurses, and ultimately results in a higher intention to leave the profession. Our results showed that nurses who worked in outpatient and ambulatory care were at decreased odds of intending to quit jobs than those who worked in an acute care setting. According to previous literature, with the COVID-19 progression, outpatient and ambulatory care practices adopted the use of telemedicine when feasible to mitigate the risk of infection, which resulted in a decline in in-person outpatient care utilization [39]. Thus, allowing for minimized nurses' risk of contamination, improved workplace safety, and possibly higher intention to remain in the profession.

Results of the current study indicated that several sociodemographic-related factors were significantly associated with the intention to quit job due to lack of satisfaction. In concordance with previous studies, females were significantly less likely to report an intention to leave their job than their male counterparts [40,41]. This is attributed to empirical evidence that females have higher job satisfaction and lower job expectations, resulting in less willingness to quit than males [42]. Also, those aged 55 years and older were significantly less likely to intend to leave their job than younger individuals, which was consistent with the previous findings [15,34]. Similarly, those who were immigrants or non-permanent residents were less likely to leave their job. A possible explanation for this finding is that the pathway to entering the healthcare workforce for immigrant nurses is more challenging and complex due to various institutional policies in place, which perhaps makes them more reluctant to quit their job than their non-immigrant counterparts [43,44]. Conversely, those with an income of under $100,000 had significantly higher odds of quitting due to lack of job satisfaction. A recent survey of registered practical nurses (RPNs) in Ontario reported that many nurses consider leaving their profession due to low wages despite the physically and mentally demanding nature of their work [23]. This finding is also in line with a study conducted in China that showed the income level of psychiatric nurses was inversely associated with the intention to leave their job [40]. Besides, nurses who lived in British Colombia were more likely to quit job than those who lived in Quebec, which could be the unintended consequence of the strict provincial order by the Government of British Columbia implemented during the early COVID-19 pandemic response that restricted healthcare workers including nurses from working at more than one site to reduce the risk of virus transmission, while such restrictions were not in place in Quebec [45,46].

Concerning health-related factors, both self-perceived general and mental health were found to be significantly associated with the intention to quit due to lack of job satisfaction. According to previous literature, elevated stress levels and lower psychological well-being were associated with higher ideation to leave job among healthcare workers [47,48]. In particular, numerous studies have reported that nurses' mental and psychological health has been adversely impacted since the emergence of the COVID-19 pandemic [12,28,42]. In line with previous studies, individuals with higher self-perceived general and mental health were less likely to intend to quit job because of lack of job satisfaction compared to participants with lower self-perceived general and mental health [48,49]. A possible explanation is that poorer mental and general health may interfere with work performance and lead to lower job satisfaction and increased professional turnover intentions. Furthermore, the highly stressful working environment of nurses leads to burnout, anxiety, and depression among nurses, resulting in a lack of satisfaction with the greater intent to leave job [50].

This is the first study to examine the impact of the mode of healthcare delivery on the nurses' intention to quit job within two years due to lack of satisfaction during the pandemic in Canada. Several limitations were identified in this study. First, given the cross-sectional nature of the study design, establishing causality may not be possible. Second, there is a possibility of selection bias as participation in this study was voluntary. Third, there might be an information bias as all responses were self-reported. Fourth, the results are subject to confounding bias such as participants' personalities, coping abilities and resilience.

Our study investigated the effect of care delivery modality on nurses' intention to quit due to lack of satisfaction in Canada. Results of this study demonstrated that nurses who provided online and blended healthcare services were significantly less likely to intend to quit job than those who offered in-person healthcare services. The findings of this study highlight the necessity of implementing effective strategies to foster nurse retention by addressing their needs and concerns at the workplace. For instance, providing opportunities for online delivery of healthcare services among nurses can allow for flexible work arrangements that could improve work-life balance, leading to higher retention rates. Another effective strategy would be increasing wages so that nurses are compensated fairly for their work, feel more satisfied with their job, and thereby more likely to stay in their profession. Implementing these measures would allow for a gratifying and positive workplace for nurses that positively influences job satisfaction, decreases turnover intention, and ultimately results in improved quality of patient care and safety during challenging events.

## Author Contributions

**Conceptualization:** Hossam Ali-Hassan.

**Formal analysis:** Safoura Zangiabadi, Hossam Ali-Hassan.

**Investigation:** Safoura Zangiabadi.

**Methodology:** Hossam Ali-Hassan.

**Project administration:** Hossam Ali-Hassan.

**Supervision:** Hossam Ali-Hassan.

**Validation:** Safoura Zangiabadi, Hossam Ali-Hassan.

**Writing – original draft:** Safoura Zangiabadi.

**Writing – review & editing:** Safoura Zangiabadi, Hossam Ali-Hassan.

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
