## [Decision Letter · Decision Letter 0]

7 Sep 2023

PGPH-D-23-01038

Effect of mode of healthcare delivery on job satisfaction and intention to quit among nurses in Canada

Dear Dr. Ali-Hassan,

Thank you for submitting your manuscript to PLOS Global Public Health. After careful consideration, we feel that it has merit but does not fully meet PLOS Global Public Health’s publication criteria as it currently stands. Therefore, we invite you to submit a revised version of the manuscript that addresses the points raised during the review process.

The reviewers have raised a number of concerns that need attention. They request revisions to the statistical analyses and revisions to improve the discussion of the context of this work, including in the title. Please take care to ensure that your conclusions are not overstated based on the results presented.

We look forward to receiving your revised manuscript.

Kind regards,

Marianne Clemence

Staff Editor

Journal Requirements:

Additional Editor Comments (if provided):

Reviewers' comments:

Reviewer's Responses to Questions

**Comments to the Author**

1. Does this manuscript meet PLOS Global Public Health’s publication criteria? Is the manuscript technically sound, and do the data support the conclusions? The manuscript must describe methodologically and ethically rigorous research with conclusions that are appropriately drawn based on the data presented.

Reviewer #1: Yes

Reviewer #2: Yes

2. Has the statistical analysis been performed appropriately and rigorously?

Reviewer #1: Yes

Reviewer #2: I don't know

3. Have the authors made all data underlying the findings in their manuscript fully available (please refer to the Data Availability Statement at the start of the manuscript PDF file)?

Reviewer #1: Yes

Reviewer #2: Yes

4. Is the manuscript presented in an intelligible fashion and written in standard English?

Reviewer #1: Yes

Reviewer #2: Yes

5. Review Comments to the Author

Reviewer #1: Very insightful and informative paper as it highlights the need for healthcare organizations to address clinical nurses' job satisfaction and challenges, particularly during pandemics, to improve continuity of care. However, the authors need to address the concerns raised, particularly the regression results (output) and the ones below:

Line 1: Improper arrangement of key words - Key words should be properly arranged in alphabetical order. Also, authors need to remove Canada as it is not a key word

Line 51: Combined years (20202023) - Authors should consider using the appropriate year

Line 63-65: Statement require at least two or more citations - Authors need to consider citing relevant references to statement

Line 84: “A study conducted in Turkey” - Authors need to identify the type of study and include in statement

Line 94: Statement of “no study” had been conducted not strong enough to justify the study - Authors to provide equally important facts to justify the study

Table 2: In the multivariate model (Regression model), It appears the OR and confidence interval values are higher than the values recorded for the adjusted odds ratio and corresponding CI for variables such as 10-19 years, multiple locations and others - Authors should critical look at the regression outcome and correct the report.

Ideally, once we don’t expect the unadjusted odds ration to be higher than the adjusted odds and the CI as well

Line 199: Outpatients - Authors can consider separating them

Line 235: Statement began with “Also” - Consider replacing with a “Again”, “Additionally”

Reviewer #2: SUMMARY OF THE RESEARCH AND YOUR OVERALL IMPRESSION

Thank you for the opportunity to review this manuscript. This manuscript describes the effect of mode of healthcare delivery on job satisfaction and intention to quit among nurses in Canada and reports nurses’ intention to quit within two years due to lack of job satisfaction. The mode of healthcare delivery was categorized into; in-person, online, or blended. I think this is a context specific and unique sample, and analyses on this sample have the potential to contribute to our understanding of in-person, online, or blended healthcare delivery and the effect it has on job satisfaction and the intention to quit during pandemics that has not been extensively studied for previous pandemics public good. My main concern is that there is no research question and or hypotheses presented.

RECOMMENDATION

1. The author should provide research question or hypotheses

2. The author should correct the year in line 51 as there seem to be a typographical error

3. The author should provide intext citation for the statement in line 232

4. The article tittle may be modified to contain pandemic or COVID-19 to make it apt. This is because the mode of healthcare delivery and intention to quit was during the Covid-19 pandemic.

IMPRESSION

The study is worthy of acceptance and publication

6. PLOS authors have the option to publish the peer review history of their article (what does this mean?). If published, this will include your full peer review and any attached files.

**Do you want your identity to be public for this peer review?** For information about this choice, including consent withdrawal, please see our Privacy Policy.

Reviewer #1: No

Reviewer #2: No

---

## [Decision Letter · Decision Letter 1]

8 Nov 2023

Effect of mode of healthcare delivery on job satisfaction and intention to quit among nurses in Canada during the COVID-19 pandemic

PGPH-D-23-01038R1

Dear Dr. Ali-Hassan,

We are pleased to inform you that your manuscript 'Effect of mode of healthcare delivery on job satisfaction and intention to quit among nurses in Canada during the COVID-19 pandemic' has been provisionally accepted for publication in PLOS Global Public Health.

Best regards,

Kaveri Mayra

Academic Editor

Reviewer Comments (if any, and for reference):

Reviewer's Responses to Questions

**Comments to the Author**

1. If the authors have adequately addressed your comments raised in a previous round of review and you feel that this manuscript is now acceptable for publication, you may indicate that here to bypass the “Comments to the Author” section, enter your conflict of interest statement in the “Confidential to Editor” section, and submit your "Accept" recommendation.

Reviewer #1: All comments have been addressed

Reviewer #2: All comments have been addressed

2. Does this manuscript meet PLOS Global Public Health’s publication criteria? Is the manuscript technically sound, and do the data support the conclusions? The manuscript must describe methodologically and ethically rigorous research with conclusions that are appropriately drawn based on the data presented.

Reviewer #1: Yes

Reviewer #2: Yes

3. Has the statistical analysis been performed appropriately and rigorously?

Reviewer #1: Yes

Reviewer #2: I don't know

4. Have the authors made all data underlying the findings in their manuscript fully available (please refer to the Data Availability Statement at the start of the manuscript PDF file)?

Reviewer #1: Yes

Reviewer #2: Yes

5. Is the manuscript presented in an intelligible fashion and written in standard English?

Reviewer #1: Yes

Reviewer #2: Yes

6. Review Comments to the Author

Reviewer #1: The authors have addressed all concerns raised in the first review and therefore accept this manuscript

Reviewer #2: The manuscript describe the effect the effect of mode of healthcare delivery on job satisfaction and intention to quit among nurses in Canada during the COVID-19 pandemic. It reported on nurses intention to quit within two years due to lack of job satisfaction. The mode of healthcare delivery considered in-person, online, or blended. I think this is a context specific and unique sample and analysis on this sample has the potential to contribute to our of in-person, online, or blended healthcare delivery and intention to quit during pandemic. This mode of healthcare delivery has not been extensively studies for previous pandemics for public good. Ethical issues were addressed appropriate as the data came from a national data set.

7. PLOS authors have the option to publish the peer review history of their article (what does this mean?). If published, this will include your full peer review and any attached files.

**Do you want your identity to be public for this peer review?** For information about this choice, including consent withdrawal, please see our Privacy Policy.

Reviewer #1: No

Reviewer #2: No
